# Deep Ensemble Learning-Based Sensor for Flotation Froth Image Recognition

**DOI:** 10.3390/s24155048

**Published:** 2024-08-04

**Authors:** Xiaojun Zhou, Yiping He

**Affiliations:** School of Automation, Central South University, Changsha 410083, China; yiping.he@csu.edu.cn

**Keywords:** flotation froth, deep ensemble learning, image recognition, membership function, TOPSIS

## Abstract

Froth flotation is a widespread and important method for mineral separation, significantly influencing the purity and quality of extracted minerals. Traditionally, workers need to control chemical dosages by observing the visual characteristics of flotation froth, but this requires considerable experience and operational skills. This paper designs a deep ensemble learning-based sensor for flotation froth image recognition to monitor actual flotation froth working conditions, so as to assist operators in facilitating chemical dosage adjustments and achieve the industrial goals of promoting concentrate grade and mineral recovery. In our approach, training and validation data on flotation froth images are partitioned in K-fold cross validation, and deep neural network (DNN) based learners are generated through pre-trained DNN models in image-enhanced training data, in order to improve their generalization and robustness. Then, a membership function utilizing the performance information of the DNN-based learners during the validation is proposed to improve the recognition accuracy of the DNN-based learners. Subsequently, a technique for order preference by similarity to an ideal solution (TOPSIS) based on the F1 score is proposed to select the most probable working condition of flotation froth images through a decision matrix composed of the DNN-based learners’ predictions via a membership function, which is adopted to optimize the combination process of deep ensemble learning. The effectiveness and superiority of the designed sensor are verified in a real industrial gold–antimony froth flotation application.

## 1. Introduction

Froth flotation is one of the most widespread and significant methods for mineral separation, applicable to nearly all types of ores. In the traditional flotation unit, skilled operators adjust chemical dosages by examining visual characteristics on the surface of the flotation froth to achieve more precise separation of minerals from impurities [1]. This not only improves the recovery of minerals, but also avoids wastage of chemicals. Therefore, the recognition of production conditions in the flotation process is of great significance for improving concentrate grade and mineral recovery. However, in actual production, relying on the naked eye to observe the froth condition on the surface of the flotation tank is costly and laborious, and manual errors can easily occur due to workers’ slightest carelessness, which can lead to frequent fluctuations in flotation generation indexes, serious loss of raw materials, high consumption of chemicals, and low resource recovery [2]. Thus, researchers began to develop machine vision-based froth flotation working condition recognition systems to automatically monitor the working condition of the flotation process [3], where the working condition is reflected by flotation froth in the flotation cell.

Machine vision is used to automatically extract visual features from flotation froth images and classify them accordingly. Since the surface state of the froth is closely related to the flotation performance index, researchers have developed some statistical distribution feature representations of flotation froth images, such as color features [4], texture features [5], bubble size [6], and a machine learning sensor can be used to classify the working condition based on the statistical distribution features of the flotation froth images. However, the traditional method is often targeted and not robust enough [7]. Deep learning is the latest progress in machine learning, which can mine deeper feature information from flotation froth images. So deep neural network (DNN) sensors are increasingly applied to the problem of flotation froth working condition recognition [8]. Incorporating state-of-the-art deep learning methods into flotation froth image recognition applications can provide performance improvement. Liu et al. [9] used AlexNet and GoogLeNet in flotation froth image recognition, and they found that the deep learning models AlexNet and GoogLeNet had higher accuracy than the local binary patterns under the same number of synthetic and real images. Zarie et al. [10] employed a convolutional neural network (CNN) sensor to classify flotation froth images captured at different air flow rates, and found that the classification accuracy of the CNN sensor was higher than that of a traditional ANN sensor, which shows the great potential of DNN in the flotation froth image classification task. With the development of transformer in machine vision [11], the ViT model has also been employed in the flotation froth image recognition [12]. It has been shown that froth image features extracted from the customized ViT have good discrimination of froth image classes.

Although deep learning sensors already have good performance in flotation froth image recognition, deeper DNN models are often considered when exploring better predictive performance. They also increase training time and the risk of overfitting of DNNs, which is not expected. At this point, deep ensemble learning is a pretty good solution that decreases the performance requirements for DNN models. In [13], a deep ensemble learning sensor is employed in sintering state recognition, and has superior recognition accuracy compared to DNNs such as VGG16, Inception-v3, ResNet50, and ResNeXt50. Zhou et al. [14] proposed an ensemble learning method based on group decision making. They categorized the working condition into eight different classes on the gold–antimony froth flotation problem, and combined the predictions of the trained LeNet, AlexNet, VGGNet, and GoogLeNet through a group decision-making method.

In this paper, a deep ensemble learning-based sensor is designed for froth flotation condition recognition, which is crucial to guide workers in the control of mineral separation dosages. The practicability and effectiveness of our method is verified on a gold–antimony flotation froth image recognition application. Specifically, pre-trained models of ViT, Swin Transformer, and EfficientNet are used to generate heterogeneous base learners in partitioned and image-enhanced training sets from collected flotation froth images. Then, DNN-based learners predict the flotation froth working condition of the images and their output goes through a membership function according to their performance on the validation set. This improves the recognition accuracy of DNN-based learners. Finally, these members construct the decision matrix and select the most probable working condition by the technique for order preference by similarity to an ideal solution (TOPSIS) based on the F1 score, which is regarded as our combination method in deep ensemble learning. Compared to other methods, our approach has the highest recognition accuracy. Therefore, we build such a deep ensemble learning-based sensor in the approach, so as to achieve industrial purpose of improving concentrate grade and mineral recovery in flotation froth image recognition.

The main contributions of this paper are summarized as follows.

(1)Heterogeneous base learners are generated with several deep learning algorithms on different training sets from images of flotation froth, which can have better diversity and accuracy.(2)Taking advantage of the DNN-based learners’ performance differences on the validation set, a membership function of the DNN-based learners’ prediction is proposed to form a decision-making problem. It also enhances the recognition accuracy of the base learners.(3)The TOPSIS method based on F1 score is put forward to select the most probable flotation froth working condition, which further improves recognition accuracy.

The rest of the study is organized as follows. In Section 2, preliminaries about deep ensemble learning are introduced. In Section 3, the proposed method is described in detail. In Section 4, experiments are conducted and the results are discussed. Finally, the conclusions are summarized in Section 5.

## 2. Preliminaries

### 2.1. Deep Neural Network

Recently, deep neural networks (DNNs) have become an important research field in artificial intelligence, and show excellent performance in many tasks [15]. As a branch of deep learning, CNNs play an important role in improving prediction performance in image classification, and significantly promote the development of DNNs. To deploy CNNs in practical applications more easily, Tan [16] proposed EfficientNet in 2019 and explored model scaling methods for convolutional neural networks (ConvNets) to achieve better performance with the lightweight network, which reduced the deployment requirements for applications. To make DNN models understand the overall structure and content of the image [17], researchers started to apply transformers to computer vision [18,19,20], since transformers can capture long-distance dependencies between pixels in the images. ViT [21] was one of the first successful studies to apply transformers to image classification. It divides and encodes an image into a set of fixed-size blocks, and then, handles these blocks through a transformer model, so as to complete image classification tasks. To achieve local sensing and deal with image information at different scales more effectively, Liu et al. [22] proposed Swin Transformer, that applied a hierarchical structure and designed a windowed self-attention mechanism on top of ViT. Finally, it achieved a good trade-off between accuracy and recognition speed. Since then, visual transformer has become an important branch of deep neural networks in image classification as well.

### 2.2. Deep Ensemble Learning

Deep ensemble learning refers to a methodology that combines multiple DNN-based learners to build a more powerful single model than its constituents, and improves the generalization ability of models [23]. The main process in deep ensemble learning is to train a few DNN-based learners, and then, fuse their outputs in a combination method.

Suppose we let ${\left\{l_{i}\right\}}_{i=1,\ldots ,m}$ denote a set of labels, K is the number of base learners, and m is the number of classes. For an observation sample x, let $P_{k}\left(y_{i}|x\right)$ be the probability that the *k*-th classifier estimates for label $y_{m}$. Then, the predictions of the base learners can be summarized as three output forms as follows [24]. In this paper, we adopt the fuzzy labels predicted by DNN-based learners for the combination process of deep ensemble learning.

Crisp label: This represents a type predicted by the base classifier with $P_{k}\left(y_{i}|x\right)\in \left\{0,1\right\}$ and $\sum_{k=1}^{K}P_{k}\left(y_{i}|x\right)=1$;Fuzzy label: This is a probabilistic explanation of the type of output of the base classifier with $\sum_{k=1}^{K}P_{k}\left(y_{i}|x\right)=1$;Possibilistic label: This relaxes the restriction of the sum of probabilities in the fuzzy label, and just needs to be greater than 0, namely, $\sum_{k=1}^{K}P_{k}\left(y_{i}|x\right)>0$.

#### 2.2.1. Generation of DNN-Based Learners

Generating a base learner is the first step in ensemble learning. Generally speaking, the better the accuracy and diversity of the base learners are, the more obviously the ensemble improves the prediction performance. Deep neural networks usually have excellent accuracy in image classification, so they can be used as base learners to effectively guarantee the initial performance of ensemble learning. As for the diversity requirements of base learners, scholars have carried out a lot of studies on the generation of base learners.

In terms of the generation of base learners, it can be categorized as either serial (boosting [25]) or parallel (bagging [26]). The base learners in the two approaches are homogeneous since a single machine learning algorithm is adopted to generate the base learners. However, the models are still structured similarly, which is not conducive to further improvement of base learner diversity. It is the same for the generation of DNN-based learners [27]. Therefore, an approach to improving the diversity of base learners further is training DNN-based learners with different structures, namely, heterogeneous base learners, where the DNN-based learner can be the classical CNN or the latest ViT based on transformer structures.

Dividing the dataset into training and testing sets is a common aspect of deep learning on datasets, where the training set is mainly provided to train the model to obtain a model with better predictive ability, while the testing set is used to evaluate the generalization performance of the model. The studies by Li et al. [28], Zheng et al. [29], and Thakkar et al. [30] adopted additional bootstrapping to partition of the dataset. This obtains different training sets to train the base learner and further expand the differences between the generated base learners. Then, the dataset on which the base learner is not chosen in the training set is used as the validation set to evaluate the weights of the base learner. Akyol adopted K-fold cross validation on an EEG dataset, splitting it into certain proportions for the training and validation sets, so that the meta-classifier could be trained on the validation set [31]. Zhou et al. [14] split a fixed proportion of one part of the dataset for training, and the other part as a validation set in order to compute the combination weights of the trained base learner. Therefore, the partition method plays an important role in the generation of base learners, and it also guides the combination process of the models.

#### 2.2.2. Combination Methods of Deep Ensemble Learning

After generating the DNN-based learners, their outputs are combined to obtain the final hypothesis in deep ensemble learning. Many combination methods for deep ensemble learning have been researched and developed in recent years, and a complete theoretical system is beginning to be formed [23,32]. The combination method affects the performance of deep ensemble learning to a certain extent, and is mainly summarized in three forms, as follows, in this paper.

(1)Methods based on meta-classifiers: To obtain the ensemble prediction results, a meta-learner can fuse base learners’ predictions and provide final results. This approach performs as a two-stage procedure in the image classification task, and is shown in Figure 1. Firstly, base learners are trained to predict target class labels. Secondly, the predictions of base learners in the validation set are taken as input of a meta-classifier, and then, the meta-classifier is trained with the input and true labels, so that it can predict the final class label accordingly. Consequently, when the deep ensemble model tests new samples, the meta-learner is able to combine the predictions of the previous DNN-based learners and obtain the final class label. In addition, the meta-learner can theoretically be any machine learning model. In this paper, the Gaussian naïve Bayes classifier is used as the meta-learner to facilitate the comparison of the prediction performance in the flotation froth image recognition application [27].(2)Methods based on voting: Voting among DNN-based learners themselves is a common combination method, shown in Figure 2. We categorize combination methods based on voting into simple majority voting and weighted majority voting according to whether additional base learner information is used or not. Simple majority voting combines base learners by calculating the number of votes of the base learners of each class label and selecting the one with the most votes as the final class label. Due to its robustness [33], many studies on ensemble learning still use it as the combination method. Simple majority voting can be regarded as using the same voting weight, but weighted majority voting assigns different weights to the base learners, which reflect the reliability of the base learners’ predictions based on their performance during training or validation. Therefore, when there is performance difference between the base learners, weighted majority voting performs better. In this paper, the weights for the base learners are based on the evaluation measures, such as accuracy and precision. In addition, the voting methods are again differentiated based on the soft and hard labels predicted by the base learners, namely, fuzzy labels and crisp labels, which are defined in Section 2.2. They can both obtain final ensemble class labels through voting rules, and simple majority voting uses hard labels, while weighted majority voting can apply any of them.(3)Methods based on aggregation rules: In this case, the process is similar to simple majority voting because there is no external information or base learner information during combination. However, instead of using the crisp label output of the base learner in simple majority voting, the fuzzy label output of the base learner is aggregated according to a rule in this approach, which is as shown in Figure 3. In this paper, we use the average, minimum, and medium aggregation methods as comparison methods, and verify the effectiveness of our proposed method in flotation froth image recognition experiments.

## 3. Methodology

### 3.1. Proposed Deep Ensemble Learning-Based Sensor

The flotation froth working condition is one of the important criteria affecting the mineral flotation process, so we recognize the working condition through the captured flotation froth images. A deep ensemble learning-based sensor is used to predict the flotation froth working condition type and guide operators to adjust the chemical dosage in order to improve the separated mineral grade. The scheme of our deep ensemble learning-based sensor is shown in Figure 4.

In the flotation froth image recognition task, we ensemble multiple heterogeneous deep neural networks trained on different training sets, and improve their prediction by a membership function and TOPSIS weighted on the F1 score. This deep ensemble learning approach is proposed to improve the generalization ability in the flotation froth application, and its main steps are as follows.

1.Categorize the flotation froth working condition into *m* class labels according to actual demand, and use experienced workers to label the flotation froth images.2.Obtain the training sets and validation sets from the images by the K-fold partitioning, so that the deep neural network can be trained on different training sets via image enhancement.3.Fine-tune the models with S kinds of pre-trained deep neural networks, then one DNN model with the highest accuracy rate on the validation set is selected to be used to form a total of K×S DNN-based learners.4.Obtain fuzzy labels of the DNN-based learners on the flotation froth testing set, and calculate the corrected prediction of the DNN-based learners through a membership function.5.Construct the decision matrix with corrected prediction and rank the working condition by TOPSIS weighted based on the F1 score. Then, select the most probable flotation froth working condition category.

The flowchart of our deep ensemble learning method for flotation froth image recognition is shown in Figure 5.

### 3.2. Generation of DNN Trained in Partitioned and Enhanced Flotation Froth Images

Training data are one of the most important factors to obtain accurate and diverse DNN-based learners. To ensure that the training data vary as much as possible, we adopt the K-fold cross validation to split the training and validation sets as in stacking, as shown in Figure 6. Specifically, there is a total of *K* subsets $\left(D_{1},\ldots ,D_{K}\right)$, all of which can be used as validation sets for the DNN-based learners. Meanwhile, the validation set is not only used to select the best DNN model in the iterative process, but also plays an important role in the membership function for the DNN model’s prediction correction, because a confusion matrix is obtained in the process of verification. Therefore, the K-fold partition to split flotation froth images is an essential part of our proposed deep ensemble learning method to improve the prediction performance of our deep ensemble learning sensor.

After the *k*-th subset $D_{k}$ is selected as the validation set, the subsets, except $D_{k}$, are processed for image enhancement (probabilistic flipping, probabilistic clipping, and weak changes in picture attributes). Then, enhanced subsets $D_{i}^{\prime }\left(i\neq k\right)$ are obtained that constitute a training set together for DNN training. This reduces the sensitivity of the base learner to the flotation froth images, thus improving the robustness of the trained DNN-based learner for the flotation froth application. In addition, the pre-trained model has extracted generic features through a wide variety of images, so using and fine-tuning it on our training data can lead to obtaining a sufficiently satisfactory DNN-based learner in fewer epochs, and the trained DNN models have better generalization ability. Therefore, our trained DNN models can also perform well when predicting an unknown testing set, even though there are probably some deviations or biases from the training set.

### 3.3. Combination Method Based on Membership Function and TOPSIS

#### 3.3.1. Design of Membership on Flotation Froth Working Condition

Assuming that $l_{i}$ is the predicted label and $p_{ks}\left(l_{i}|x\right)$ is the fuzzy label of the test sample *x* from the DNN-based learner $T_{ks}$, the confusion matrix $CM_{ks}$ is obtained on the validation set $D_{k}$ from DNN-based learner $T_{ks}$.
(1)$$CM_{ks}=\left[\begin{array}{ccc} L_{11} & \cdots & L_{1m}\\ \vdots & \ddots & \vdots \\ L_{m1} & \cdots & L_{mm} \end{array}\right]$$
where *m* is the number of flotation froth class labels, $T_{ks}$ represents the *s*-th pre-trained DNN model, and the DNN is validated on the *k*-th subset as the validation set.

Given that $L_{ji}$ is the number of flotation froth images in the validation set when the true label is $l_{j}$ and simultaneously the predicted label is $l_{i}$, then its prior frequency is calculated as follows:(2)$$p_{ks}\left({\hat{l}}_{j}|l_{i}\right)=L_{ji}/\sum_{k=1}^{m}L_{ki}$$
where $p_{ks}\left(l_{i}|x\right)$ is the fuzzy labels predicted by the DNN-based learner $T_{ks}$.

If there is a sufficiently large number of samples in the validation set, we can regard the frequency as a probability, and then, use a membership function on fuzzy labels of this DNN-based learner in Equation (Equation 3).
(3)$$\begin{array}{cc} p_{ks}\left({\hat{l}}_{j}|x\right) & =f_{membership}\left(p_{ks}\left(l_{1}|x\right),p_{ks}\left(l_{2}|x\right)\ldots p_{ks}\left(l_{n}|x\right)\right)\\ \; & =\sum_{i=1}^{n}p_{ks}\left({\hat{l}}_{j}|l_{i}\right)p_{ks}\left(l_{i}|x\right) \end{array}$$

The computation of the membership function on fuzzy labels in this paper draws on the advantage of the weight in the weighted voting method and becoming more flexible. It not only utilizes all the performance information of the DNN-based learners’ confusion matrix, but also considers the original fuzzy label potential links between different categories, in order to represent a better prediction of every DNN-based learner. Finally, the fuzzy labels, via the membership function, construct a decision matrix in Equation (Equation 4), which can be solved by means of decision theory.
(4)$$X_{m\times B}=\left[\begin{array}{ccc} x_{11} & \cdots & x_{1B}\\ \vdots & \ddots & \vdots \\ x_{m1} & \cdots & x_{1B} \end{array}\right]=\left[\begin{array}{ccc} p_{1}\left({\hat{l}}_{1}|x\right) & \cdots & p_{B}\left({\hat{l}}_{1}|x\right)\\ \vdots & \ddots & \vdots \\ p_{1}\left({\hat{l}}_{m}|x\right) & \cdots & p_{B}\left({\hat{l}}_{m}|x\right) \end{array}\right]$$
where *B* is the number of DNN-based learners and B=K×S.

#### 3.3.2. Decision-Making Method for Flotation Froth Working Condition

TOPSIS is a decision-making method commonly used in the decision-making field, and its core idea is the distance between each solution and the positive and negative ideal solutions [34]. We sort the working condition by calculating the proximity of each category to the positive and negative ideal solution according to the membership decision matrix. The closest one has the largest evaluation value and is determined as the final predicted working condition.

The steps of our proposed ranking method based on TOPSIS weighted by the F1 score of the DNN-based learners are as follows.

1.Fuzzy labels, via the membership function, can constitute the decision matrix $X_{m\times t}$, which is shown in Equation (Equation 4).2.Compute the positive ideal solution $C_{i}^{+}$ and the negative ideal solution $C_{i}^{-}$:
(5)$$C_{i}^{+}=\max_{j=1,\ldots ,KS}\left(x_{ij}\right)$$
(6)$$C_{i}^{-}=\min_{j=1,\ldots ,KS}\left(x_{ij}\right)$$3.Calculate of the distance from the working condition $l_{i}$ to the positive and negative ideal solutions, taking the F1 score for each category of the DNN-based learner as weights.
(7)$$s_{i}^{+}=\sqrt{{\sum_{j=1}^{KS}F_{ij}\left(c_{ij}-c_{i}^{+}\right)}^{2}}$$
(8)$$s_{i}^{-}=\sqrt{{\sum_{j=1}^{KS}F_{ij}\left(c_{ij}-c_{i}^{-}\right)}^{2}}$$
where $F_{ij}$ represents the F1 score of DNN-based learner j in the flotation froth working condition $l_{i}$.4.Calculate the value of the ranking indicator $f_{i}$ for each working condition $l_{i}$.
(9)$$f_{i}=\frac{s_{i}^{-}}{s_{i}^{-}+s_{i}^{+}}$$5.Let $i_{o}=\underset{i}{\arg\max f_{i}}$, and choose $l_{i_{o}}$ as the final working condition label.

## 4. Experimental Results

### 4.1. Data Description

Gold–antimony froth flotation is an important process that utilizes the principle of bubble buoyancy to separate gold–antimony material from ore slurry [35]. In this paper, gold–antimony froth flotation images were collected to validate the effectiveness of the proposed method. We set up a computer, lamp, camera, and other devices in a gold–antimony flotation industry in Hunan, China, and then, collected flotation froth images from the flotation tanks for experiments. Workers experienced with flotation froth evaluated the working condition according to the characteristics of the images, and categorized them into eight different types, which are shown in Figure 7. These categories reflect the gold–antimony production condition so as to assist in adjusting the chemical dosages, which can improve the recovery of minerals and reduce the cost. Each category is uniformly distributed in order to verify the effectiveness of the method proposed in this paper.

In addition, each captured flotation froth image is 800 px × 600 px and clear enough to present its working condition in Figure 7. There are a total of 2400 flotation froth images and a testing set of 480 images is split off. The rest of the images are used for DNN training and verification.

### 4.2. Experiment Setting

In our experiment, the pre-trained EfficientNet [16], ViT [21], and Swin Transformer [22] were each used to train a total of 12 heterogeneous DNN-based learners on 4-fold-partitioned flotation froth data, and the trained DNN-based learners were used as our deep ensemble learning sensor. The 4-fold partitioning of the flotation froth data resulted in a quarter of the images being used as the validation set, which could sufficiently represent unknown flotation froth samples and ensure the base learners’ generalization ability. In addition, our training set was flipped probabilistically, clipped probabilistically, rotated probabilistically, and jittered in color attributes to enhance the flotation froth images.

The deep neural network settings for training are shown in Table 1. As is shown in Table 1, we attempted to set the same parameters in different DNNs, such as epoch and batch size. Then, we fine-tuned these pre-trained DNNs, and the DNN with the highest accuracy in the validation set in its training epoch process was taken as the DNN-based learner for the deep ensemble learning-based sensor.

### 4.3. Comparison with Base Learners

The recognition accuracy of 12 DNN-based learners for the gold–antimony froth flotation application is shown in Figure 8. We can find that it is the same in the validation and testing set of each base learner, which indicates that the data distribution is basically consistent. In addition, the accuracy of the base learners mostly reaches 90% on the validation and testing sets. This illustrates that our K-fold partitioning and image enhancement are useful for training good DNN-based learners.

In Figure 8, the symbols of the DNN-based learner signify which deep learning network was trained under which fold in the flotation froth data. For example, ViT1/ SwinT1/ EfficientNet1 indicates that ViT/ Swin Transformer/ EfficientNet is evaluated in first fold of the validation set; after, it is trained on the rest of the subsets. Other DNN-based learners trained on other subsets are also represented in the same way. The best base learner performance on the validation set is EfficientNet3, with the accuracy of 94.38%. However, if the base learner with the highest accuracy on the validation set is selected to be used in the testing set, it has a lower recognition accuracy of 93.75% compared to the other DNN-based learners, one of which achieves 96.67% accuracy in the testing set. This shows that if one of these base learners is selected for the application of gold–antimony froth flotation condition recognition, it is difficult to pick out a DNN with sufficiently good generalization ability according to its performance on the validation set, because an overfitted DNN-based learner is easily chosen, which decreases the recognition accuracy on the testing set.

Therefore, we combine these DNN-based learners using deep ensemble learning, and then, apply the ensemble model to the testing set of gold–antimony froth flotation image recognition. In our method, the membership of the flotation froth working condition constructs the decision matrix and TOPSIS weighted by the F1 score is applied to combine the base learners; its confusion matrix is shown in Figure 9. According to Figure 9, there are only a total of eight misclassified flotation froth images, and we can calculate that the accuracy of our proposed deep ensemble learning approach reaches 98.33%, which is better than any of the base learners. This demonstrates that our deep ensemble learning-based sensor improves the generalization ability of the model and verifies the effectiveness of our method.

### 4.4. Comparison with Classic Ensemble Learning Methods

We compare the proposed method with classic ensemble learning methods such as Adaboost [36], gradient boosting [37], bagging [26], XGBoost [38], and random forest [39]. The inputs in the classical ensemble learning method are 38 statistical features extracted from the froth flotation images referring to [40], including the mean of gray, bubble size distribution, length–width ratio, and so on. Finally, the experimental results are shown in Figure 10.

According to Figure 10, the recognition accuracy of the classical ensemble learning method on the gold–antimony froth flotation testing set is not more than 90%, which is even lower than that of the DNN-based learner shown in Figure 8, indicating that the DNN is more suitable for the recognition task. Classic ensemble learning does not perform well because the extracted statistical features may lose some important information about the flotation froth image or mix with a lot of redundant information, which results in worse generalization ability of the trained ensemble model. This is also an important reason why most studies prefer deep learning methods in image classification.

### 4.5. Comparison with Other Combination Methods

In order to further verify the superiority of our proposed method in flotation froth image recognition applications, we compare our approach with other combination methods in deep ensemble learning. Specifically, the minimum (MIN), product (PRO), and average (AVR) aggregation rules [41], the soft voting method (SV) [42] and hard voting method (HV) [42], weighted hard voting based on $F_{in}$ (FHV) [14], weighted voting using global accuracy (WVGA) [27], weighted voting using class precision (WVCP) [27], Bayesian meta-classifier using predictions (PBAY) [27], and multilayer perceptron meta-classifier using predictions (PMLP) [43] are taken as combination methods on the testing set and the experimental results are shown in Figure 11. As shown in Figure 11, all of these combination methods clearly improve the flotation froth image recognition accuracy on the testing set compared to the DNN-based learner in Figure 8, which demonstrates the better generalization performance of deep ensemble learning. In addition, the combination method based on the aggregation rules essentially outperforms the voting method, implying that the aggregation rules work well in this application. The stacking method using a Bayesian classifier and MLP as the meta-learner has poor performance in this application, because the meta-learner, with a complicated structure, produces an overfitting phenomenon due to the quite good diversity and accuracy of the DNN-based learners. Therefore, the relatively simpler combination method has better performance. Last, but not least, our approach has the highest recognition accuracy, which verifies the superiority of our combination method in the application.

To further illustrate the improvement in our combination method, we display the accuracy of the DNN-based learners on the testing set when using the membership function or not in Figure 12. According to Figure 12, most of the 12 DNN-based learners via the membership function have higher accuracy on the testing set, verifying that the membership function improves the prediction performance of the base learners.

In addition, we analyzed a sample of gold–antimony flotation froth images in the testing set for observation. It is hard to be categorized into “good”, but its real working condition type is “good”, because the DNN-based learners prefer categorizing it as “fair”. The fuzzy labels of the DNN-based learners are shown in Table 2, where the bold number is the largest prediction. After the DNN-based learners’ predictions are put through Equation (Equation 3), their predictions are closer to the real working condition, according to Table 3. It can be found that most of the base learners do not regard it to be “good”, and thus, cannot correctly predict its working condition even in the soft voting combination method. Afterwards, the DNN-based learners’ predictions are put through Equation (Equation 3), and their predictions are closer to the real working condition. Then, the TOPSIS method, weighted by the F1 score, finally makes the evaluation value of the right type, “good”, larger than that of the wrong type, “fair”, and thus, the right flotation froth working condition can be determined successfully.

Finally, we carried out certain image attribute changes and rotations randomly on the flotation froth testing data, which simulate the image interference in the real flotation froth process. The results are shown in Figure 13. It can be found that the accuracy of our proposed method does not decrease obviously and reaches 96.88%, while the accuracy of the other combination methods is not better than it, which verifies the robustness of our method.

## 5. Conclusions

In this study, a deep ensemble learning-based sensor is designed to automatically monitor the flotation working condition on the basis of the captured flotation froth images, so that the chemical dosages can be adjusted accordingly. The K-fold partition is used in partitioning training and validation data and image enhancement is performed on the training data to generate DNN-based learners with diversity and accuracy in different deep learning algorithms. With the predictions obtained from these DNN-based learners, a membership function utilizing the performance information of the DNN-based learners during the validation is proposed to improve the recognition accuracy of the DNN-based learners. Then, membership of the flotation froth working condition constructs the decision matrix and TOPSIS weighted by the F1 metric is proposed to combine the base learners. Finally, the most probable working condition is selected to assist operators in adjusting the chemical dosages.

The experimental results show that the sensor based on deep ensemble learning method can effectively recognize the flotation froth working condition and achieves the accuracy of 98.33% on the testing data of the gold–antimony flotation froth application, which demonstrates the superiority of the proposed approach compared to other common methods.

In future work, we will explore different DNN models and more performance metrics of base learners, as well as investigate more deep ensemble learning based on other decision-making methods to improve the applicability and interpretability of our approaches. Moreover, different flotation froth applications under diverse industrial conditions will be explored to enhance the generalizability of our approach, especially in variable flotation froth working conditions. In addition, we also believe that this work has potential to be applied to other recognition tasks like regression and object detection.

## Figures and Tables

**Figure 1 sensors-24-05048-f001:**
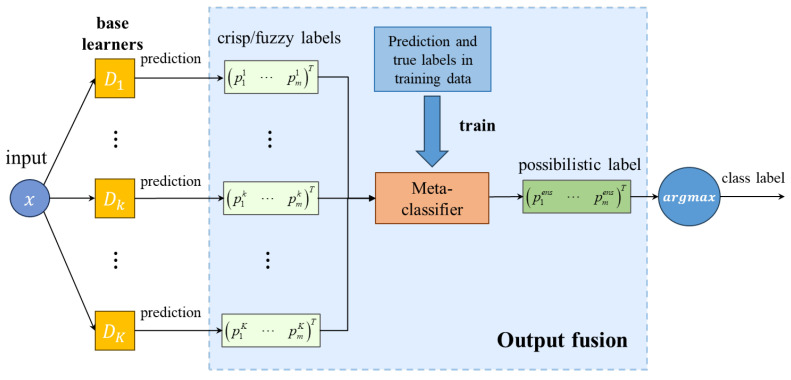
Combination methods based on meta-classifiers.

**Figure 2 sensors-24-05048-f002:**
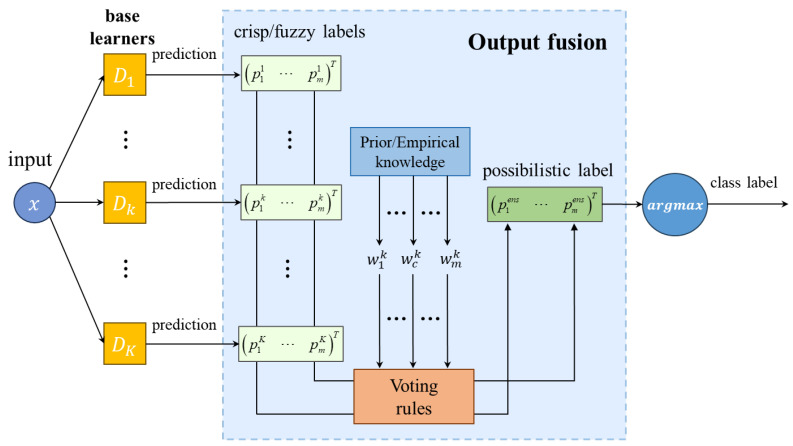
Combination methods based on voting.

**Figure 3 sensors-24-05048-f003:**
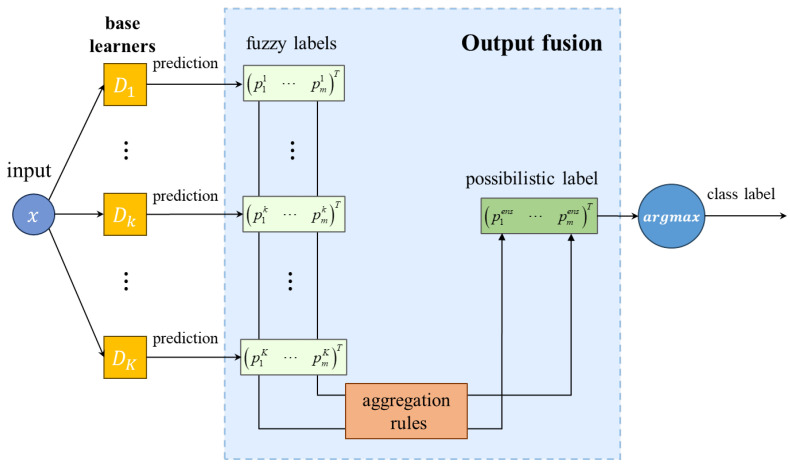
Combination methods based on aggregation rules.

**Figure 4 sensors-24-05048-f004:**
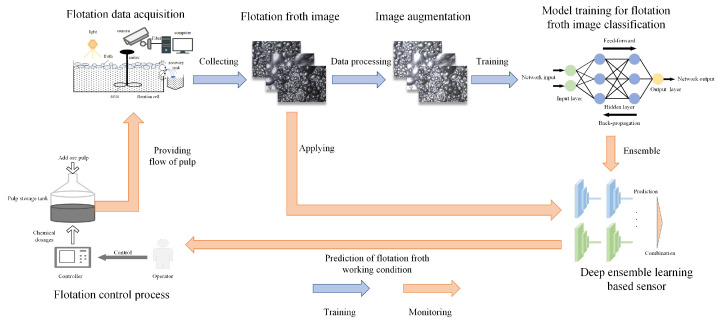
Scheme of flotation froth image recognition in our deep ensemble learning-based sensor.

**Figure 5 sensors-24-05048-f005:**
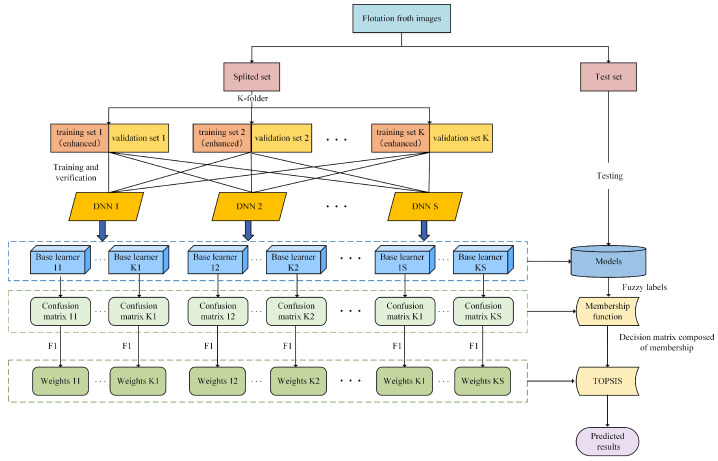
The flowchart of our deep ensemble learning method for flotation froth image recognition.

**Figure 6 sensors-24-05048-f006:**
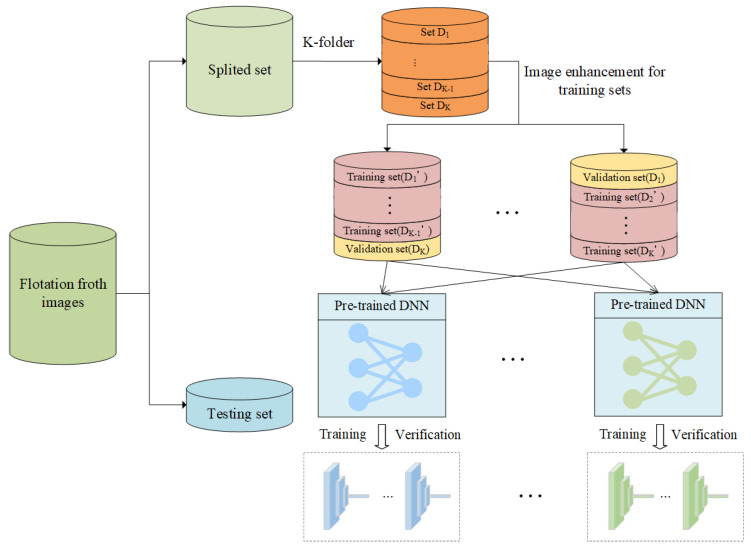
Partitioning of training, validation, and testing sets.

**Figure 7 sensors-24-05048-f007:**
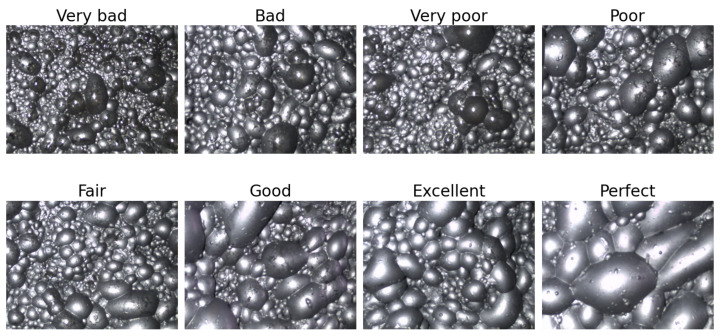
Flotation froth under eight different working conditions.

**Figure 8 sensors-24-05048-f008:**
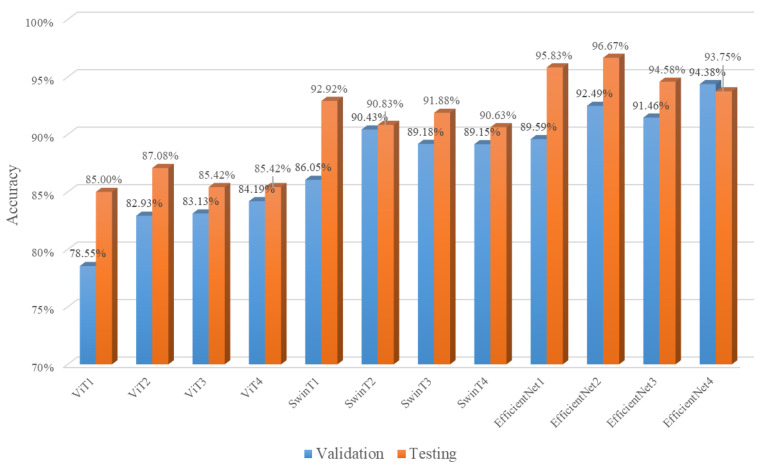
Accuracy in base learners in flotation froth image recognition task.

**Figure 9 sensors-24-05048-f009:**
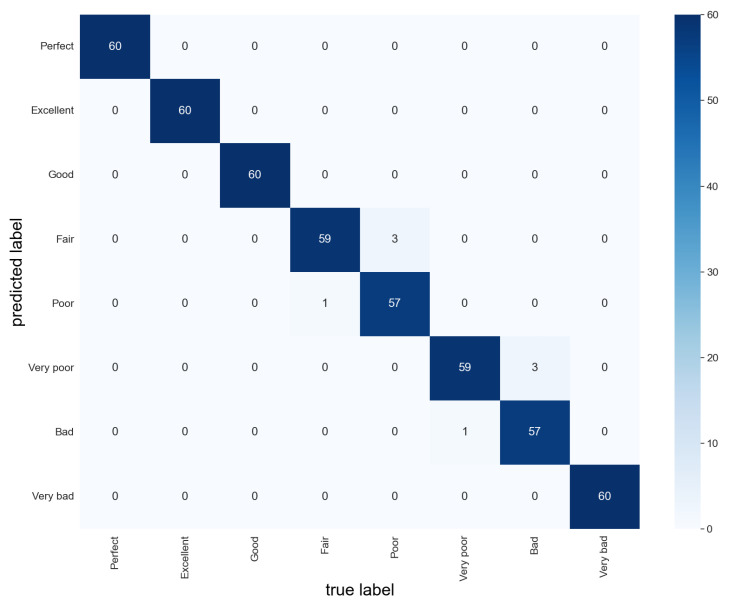
Confusion matrix of our method.

**Figure 10 sensors-24-05048-f010:**
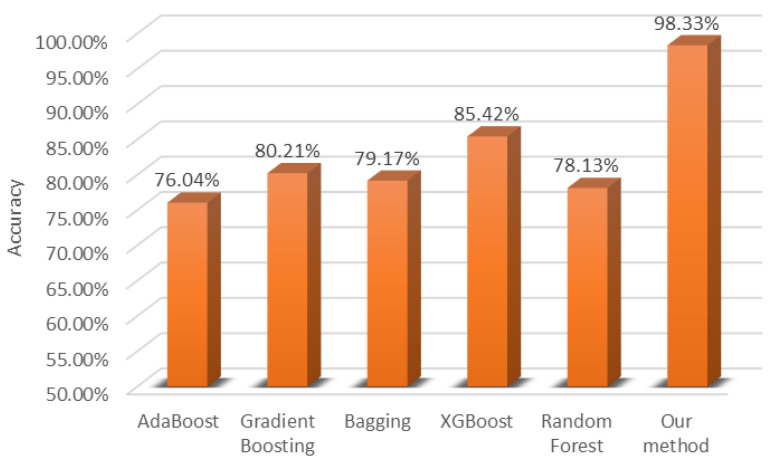
Accuracy of classical ensemble learning.

**Figure 11 sensors-24-05048-f011:**
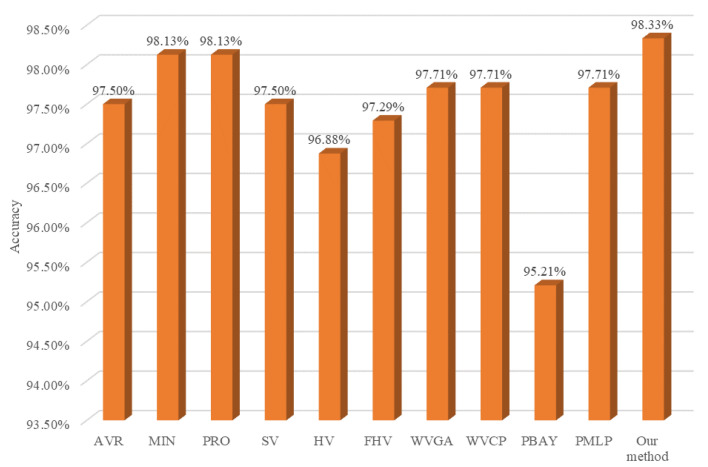
Accuracy of different combination methods on testing data.

**Figure 12 sensors-24-05048-f012:**
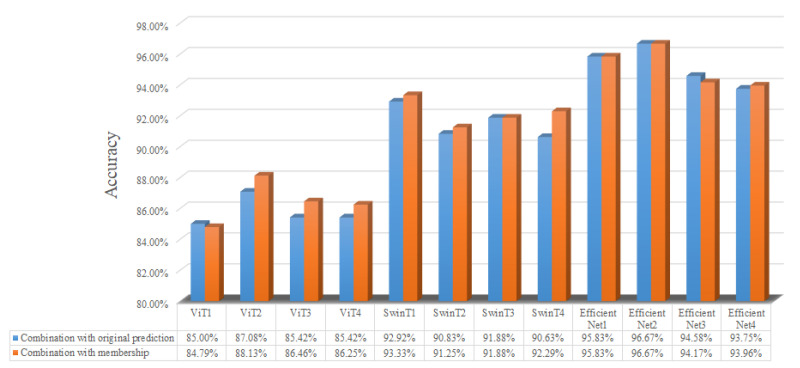
Accuracy of base learners using membership function or not.

**Figure 13 sensors-24-05048-f013:**
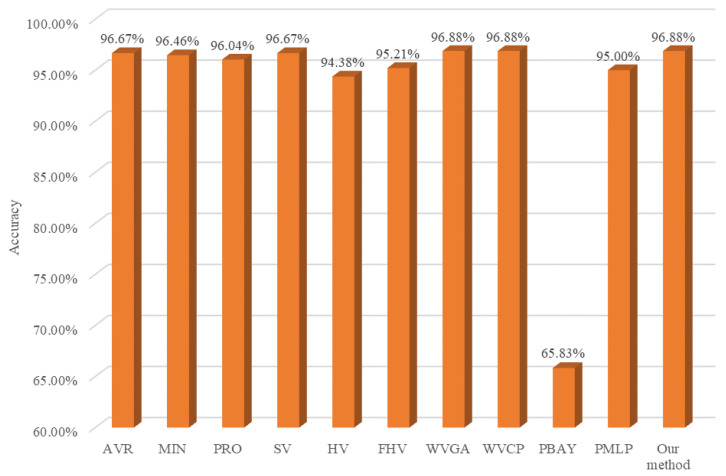
Accuracy of different combination methods on testing data under image interference.

**Table 1 sensors-24-05048-t001:** Experimental settings of DNN.

DNN	EfficientNet [16]	ViT [21]	Swin Transformer [22]
Epoch	50	50	50
Batch size	20	20	20
Optimization function	Adam	Adam	AdamW
Input dimension	224 × 224	224 × 224	224 × 224
Model	efficientnet_v2_s	vit_b_16	swin_tiny_patch4_window7_224
Last layer	Linear(1280,8)	Linear(768,8)	Linear(768,8)

**Table 2 sensors-24-05048-t002:** Original fuzzy labels of base learners.

Type	ViT1	ViT2	ViT3	ViT4	SwinT1	SwinT2	SwinT3	SwinT4	EfficientNet1	EfficientNet2	EfficientNet3	EfficientNet4	AVR
Very bad	0.001	0.000	0.001	0.001	0.000	0.000	0.000	0.000	0.000	0.000	0.001	0.000	0.000
Bad	0.000	0.001	0.000	0.000	0.000	0.000	0.000	0.000	0.000	0.000	0.000	0.000	0.000
Very poor	0.000	0.000	0.000	0.000	0.000	0.000	0.000	0.000	0.000	0.000	0.000	0.000	0.000
Poor	0.011	0.149	0.038	0.075	0.000	0.000	0.001	0.047	0.001	0.006	0.005	0.000	0.028
Fair	0.125	**0.821**	0.146	**0.666**	**0.983**	0.001	**0.659**	**0.941**	0.034	**0.933**	0.075	0.006	**0.449**
Good	**0.832**	0.012	**0.714**	0.044	0.015	**0.997**	0.009	0.011	**0.770**	0.058	0.006	**0.994**	0.372
Excellent	0.027	0.017	0.089	0.210	0.002	0.000	0.332	0.001	0.194	0.002	**0.912**	0.000	0.149
Perfect	0.003	0.001	0.012	0.004	0.000	0.001	0.000	0.000	0.000	0.001	0.000	0.000	0.002

**Table 3 sensors-24-05048-t003:** Fuzzy labels of base learners via membership function.

Type	ViT1	ViT2	ViT3	ViT4	SwinT1	SwinT2	SwinT3	SwinT4	EfficientNet1	EfficientNet2	EfficientNet3	EfficientNet4	Our Method
Very bad	0.001	0.000	0.001	0.001	0.000	0.000	0.000	0.000	0.000	0.000	0.001	0.000	0.000
Bad	0.000	0.014	0.010	0.017	0.000	0.000	0.000	0.007	0.000	0.000	0.000	0.000	0.000
Very poor	0.001	0.000	0.000	0.000	0.011	0.000	0.000	0.000	0.000	0.000	0.000	0.000	0.000
Poor	0.042	0.191	0.044	0.251	0.249	0.016	0.088	0.053	0.005	0.090	0.013	0.017	0.027
Fair	0.081	**0.669**	0.122	**0.441**	**0.656**	0.050	**0.572**	**0.929**	0.031	**0.792**	0.082	0.006	0.447
Good	**0.756**	0.052	**0.628**	0.089	0.082	**0.884**	0.008	0.011	**0.733**	0.052	0.006	**0.977**	**0.491**
Excellent	0.026	0.069	0.097	0.188	0.002	0.016	0.332	0.001	0.188	0.059	**0.897**	0.000	0.166
Perfect	0.092	0.004	0.098	0.013	0.000	0.033	0.000	0.000	0.044	0.006	0.000	0.000	0.003

## Data Availability

Due to privacy or ethical restrictions data are unavailable.

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
