# Peer review of "Deep Ensemble Learning-Based Sensor for Flotation Froth Image Recognition"

_sensors, 2024, doi:10.3390/s24155048_

Round 1
Reviewer 1 Report
Comments and Suggestions for Authors
This manuscript proposes a deep ensemble learning method based sensor for flotation froth image recognition. The experimental results look very good. However, some small issues in this manuscript should be addressed.
- In line 256, 'pks(li|x) is the fuzzy prediction label', which seems not correct, 'because li is the predicted label'.
- Because Figure 10 is referenced first, followed by Figure 9, the order of the two figures should be swapped to ensure that their appearance is consistent with the order of their reference.
- Abbreviations in tables and figures should be included throughout the text or the authors provide literature references when these Abbreviations first appear, such as AVR (in Table 2) and MCTF (in Table 3).
- 'McSV' (in line 399) and 'MCSV' (in Table 3) should be the same abbreviations.
- Please explain why this method only applies to the recognition of image flotation froth image.
More details:
The method proposed in this manuscript is feasible, and the experimental results are very good, so I gave the conclusion ,minor revision. The text of the manuscript that needs to be modified includes at least five specific grammar and writing issues.
Of course, I also hope the author can carefully proofread the entire text again. Have a nice day! Comments on the Quality of English LanguageMinor editing of English language required.
Author Response
Please see the attachment. Note our changes made in the revision are highlighted in "red", and responses to comments are highlighted in "blue".

Reviewer 2 Report
Comments and Suggestions for Authors
The paper presents a deep ensemble learning-based sensor designed for flotation froth image recognition. This sensor aims to assist operators in adjusting chemical dosages during the mineral separation process by providing accurate recognition of flotation froth working conditions. The authors use a combination of pre-trained deep neural networks (ViT, Swin Transformer, and EfficientNet) and propose a membership function along with TOPSIS based on F1 score to enhance the recognition accuracy. The methodology is validated with real industrial data from a gold-antimony flotation application, showing significant improvements over traditional methods.
The manuscript is well-written, and the research is relevant to the readers of Sensors. However, here are a few recommendations for potential enhancements and considerations that could further strengthen your manuscript:
(1) Please complete ‘Author Contributions’ section.
(2) The manuscript primarily validates the proposed method using a dataset from a single industrial application (gold-antimony froth flotation). This limits the generalizability of the findings. Please consider testing on multiple datasets from different types of mineral processing plants. This would strengthen the method’s applicability across various scenarios.
(3) While the paper demonstrates high accuracy, there is limited discussion on the real-time implementation of the sensor. The latency and processing time required for real-time froth image recognition and decision-making are crucial for practical industrial applications.
(4) The paper does not thoroughly address the potential for overfitting, especially given the complex models used and the relatively small size of the dataset. Additional techniques to prevent overfitting, such as dropout or data augmentation, could be explored and discussed.
(5) Although the paper compares the proposed method with classical ensemble learning techniques, it lacks a detailed comparison with other state-of-the-art machine learning methods, such as advanced convolutional neural networks (e.g., ResNet, DenseNet) and transformer-based models in similar applications.
(6) The scalability of the proposed method to larger datasets or more diverse industrial conditions is not discussed. The ability of the model to maintain performance with an increasing number of classes or more complex froth images needs to be evaluated.
(7) The robustness of the proposed sensor to noisy or low-quality images, which are common in real-world industrial environments, is not thoroughly investigated. Further analysis on how the sensor performs under suboptimal imaging conditions would be beneficial.
(8) The method’s sensitivity to environmental changes, such as lighting conditions, camera angles, and variations in froth appearance due to different ore properties, is not explored. Such variability could impact the sensor’s performance in practical applications.
Author Response

(The authors gave the same response as above.)

Round 2
Reviewer 2 Report
Comments and Suggestions for Authors
All questions have been answered. No further questions.